# Risk Factors of Daptomycin-Induced Eosinophilic Pneumonia in a Population with Osteoarticular Infection

**DOI:** 10.3390/antibiotics10040446

**Published:** 2021-04-16

**Authors:** Laura Soldevila-Boixader, Bernat Villanueva, Marta Ulldemolins, Eva Benavent, Ariadna Padulles, Alba Ribera, Irene Borras, Javier Ariza, Oscar Murillo

**Affiliations:** 1Infectious Diseases Service, IDIBELL-Hospital Universitari Bellvitge, Feixa Llarga s/n, Hospitalet de Llobregat, 08907 Barcelona, Spain; laura.soldevila@bellvitgehospital.cat (L.S.-B.); bvillanueva@bellvitgehospital.cat (B.V.); mulldemolins@bellvitgehospital.cat (M.U.); eva.benavent@bellvitgehospital.cat (E.B.); ARibera@scias.com (A.R.); Iborras@scias.com (I.B.); jariza@bellvitgehospital.cat (J.A.); 2Bone and Joint Infection Study Group of the Spanish Society of Clinical Microbiology and Infectious Diseases (GEIO-SEIMC), 28003 Madrid, Spain; 3Pharmacy Department, IDIBELL-Hospital Universitari Bellvitge, Feixa Llarga s/n, Hospitalet de Llobregat, 08907 Barcelona, Spain; apadulles@bellvitgehospital.cat; 4Spanish Network for Research in Infectious Diseases (REIPI RD16/0016/0003), Instituto de Salud Carlos III, 28029 Madrid, Spain

**Keywords:** daptomycin, eosinophilic pneumonia, risk factors

## Abstract

Background: Daptomycin-induced eosinophilic pneumonia (DEP) is a rare but severe adverse effect and the risk factors are unknown. The aim of this study was to determine risk factors for DEP. Methods: A retrospective cohort study was performed at the Bone and Joint Infection Unit of the Hospital Universitari Bellvitge (January 2014–December 2018). To identify risk factors for DEP, cases were divided into two groups: those who developed DEP and those without DEP. Results: Among the whole cohort (*n* = 229) we identified 11 DEP cases (4.8%) and this percentage almost doubled in the subgroup of patients ≥70 years (8.1%). The risk factors for DEP were age ≥70 years (HR 10.19, 95%CI 1.28–80.93), therapy >14 days (7.71, 1.98–30.09) and total cumulative dose of daptomycin ≥10 g (5.30, 1.14–24.66). Conclusions: Clinicians should monitor cumulative daptomycin dosage to minimize DEP risk, and be cautious particularly in older patients when the total dose of daptomycin exceeds 10 g.

## 1. Introduction

Daptomycin is a cyclic lipopeptide antibiotic approved for use against complicated skin and soft tissue infection, *Staphylococcus aureus* bacteremia and right-sided infective endocarditis. However, daptomycin has become widely used also in staphylococcal osteoarticular infections because of its remarkable anti-biofilm activity. Indeed, current guidelines advise for its use mainly as an initial induction course of intravenous antimicrobial therapy and often in combination with other antibiotics to avoid the appearance of resistance [1,2]. In this setting, the use of daptomycin for prolonged periods should be balanced between the potential benefits in the outcome and the risk of adverse events [3,4,5].

Although daptomycin has proven safety, daptomycin-induced eosinophilic pneumonia (DEP) is a rare but severe adverse effect [6,7]. This toxicity is partially related to the usual daptomycin uptake by pulmonary surfactant in the alveoli, which may lead to concentrations high enough to cause injury but also to impair its efficacy; in fact, daptomycin is not recommended to treat pulmonary infections. Despite the fact that the pathophysiology is not totally clear, it seems that DEP is an antigen-mediated process in which alveolar macrophages and T-cells may be activated, which then release interleukin-5 that causes eosinophil production and migration to the lungs. Additionally, alveolar macrophages can also excrete cytokines that selectively recruits eosinophils, which may promote further eosinophil accumulation into the lungs [8,9].

Since the introduction of daptomycin, while some cases of DEP have been reported, these have only described the most common clinical manifestations and outcome [10,11,12,13]. To date, therefore, we do not know which factors are associated with DEP and thus, in the present study we aimed to determine the risk factors for developing DEP.

## 2. Results

In total, 229 cases received at least one dose of daptomycin and among them, 11 (4.8%) had DEP; a comparison of both groups in regard with main clinical and analytical characteristics is presented in Table 1. All DEP cases underwent a chest X-ray while on daptomycin therapy, which showed peripheral lung infiltrates (alveolar or interstitial), and only one patient had a CT scan that showed radiological findings of organizing pneumonia. In contrast, only 26% cases (57/218) of the remaining cohort underwent a chest X-ray, which was considered similar to the baseline one. Of interest, the performance of a chest X-ray significantly increased in accordance with the length of daptomycin therapy, ranging from 21% in cases treated less than 7 days to 42% in those treated more than 14 days (*p* = 0.005). With regard to the age of patients, cases aged ≥70 years underwent a chest X-ray during daptomycin therapy in greater proportion than younger patients (31% vs. 24%, respectively).

All DEP cases were treated with daptomycin withdrawal and seven (64%) with corticosteroid therapy. One patient, who had a delay in diagnosis of DEP and therapy, died because of respiratory failure.

In the univariate analysis (Table 1), factors associated with DEP were advanced age, the presence of comorbidities measured by Charlson score, long treatment with daptomycin and high values of TCDD. Concisely, daptomycin therapy for two weeks or longer was associated with high risk of DEP (HR 7.71, 95%CI 1.98–30.09), as well as TCDD values ≥10 g (HR 5.30, 95%CI 1.14–24.66). The presence of blood eosinophilia at the end of daptomycin treatment was significantly higher in DEP cases than in controls (82% and 16%, respectively; *p* < 0.001), as well as leucocyte counts and C-reactive protein values were also higher in DEP cases.

We noted that among older patients aged ≥70 years (*n* = 123), the percentage with DEP (8.1%; 10/123) almost doubled the value of the whole cohort. Also, the percentage of cases with DEP increased significantly among cases aged ≥70 years in comparison with the whole cohort either in cases treated for >14 days or in those with high values of TCDD (Figure 1).

Finally, 25 cases had a re-challenge to daptomycin therapy, and two of these presented promptly with DEP (8%) by 4 and 8 days after the re-challenge (3 and 5 months after the first exposure, respectively). Both cases presented blood eosinophilia after the first course of treatment, having received >11 g over >14 days. By contrast, among the remaining patients re-challenged with daptomycin, the eosinophilia was only observed at the prior exposure for four patients (17%).

## 3. Discussion

In the present study we reported the main risk factors for developing DEP in a population with osteoarticular infections, providing important new information that may be helpful to clinicians.

Daptomycin has been reported as the leading cause of drug-induced eosinophilic pneumonia [14], and clinicians should maintain a high index of suspicion for DEP because of its potential severity. Although most of cases in our series were resolved by daptomycin withdrawal and corticosteroid therapy, one patient died, which illustrates the inherent risk of failing to identify DEP promptly.

Daptomycin use for the treatment of osteoarticular infection is currently recommended mainly against staphylococcal infections and as initial induction antimicrobial therapy [1,2]. In contrast with the high activity of daptomycin in animal and in vitro studies, its clinical efficacy reported from non-comparative studies appeared to be quite similar to other therapies [5,15,16]. However, prolonged therapy at higher doses than usual seems to be increased in recent years. Our cases were treated with daptomycin for a median of 19 days and resulted in DEP proportions of 4.8% overall and 8.1% among those aged ≥70 years. This data may seem high compared with previous experiences and without placing it in context. Thus, populations with 102 cases of infective endocarditis and 43 cases of complex osteoarticular infections that received high doses of daptomycin (median 8.2 mg/kg/d) for long periods (20–80 days), the authors showed 3% and 4.6% developed DEP, respectively [17,18]. These are consistent with our results given that the populations in both studies were younger (mean age 61.5 years) than in the present study.

Taking all the previous into account, the prolonged therapy with daptomycin against osteoarticular infections or the use of higher doses than usual should be considered on the basis of a list of pros and cons. Probably, the efficacy of daptomycin therapy is related with its anti-biofilm activity and can be benefited through an initial intensive phase of treatment (i.e., 7–14 days). Further therapy should be balanced with inconveniences derived from its use; indeed, monitoring for daptomycin toxicity appears crucial in long therapies and includes not only the risk for DEP but also other adverse events such as rhabdomyolysis. In our experience, performance of chest-X ray was useful to identify DEP and thus, it appears as valid screening to be interpreted together with other clinical signs and analytical parameters.

To our knowledge, no previous studies had been performed to analyze the risk factors of developing DEP. We identified advanced age, high values of Charlson comorbidity index, length of daptomycin therapy and TCDD as the main risk factors for DEP. Of interest, we show that patients older than 70 years, which commonly have more underlying diseases, are at higher risk of DEP; however, further research is needed to evaluate the importance of particular comorbidities in increasing the risk of DEP.

Regarding cumulative dosages of daptomycin and long therapies, our results seem to be consistent with previous works. Hirai et al. [19] reported 40 cases of DEP, 73% of them received a daptomycin dosage >6 mg/kg/d for a median of 14.8 days, whereas the remaining cases were treated with daptomycin at ≤6 mg/kg/d for a median of 23 days. In a systematic review of DEP cases the mean length of daptomycin therapy was 2.8 weeks and main indication for treatment was osteoarticular infection [20]. Overall, it seems that higher risk of DEP is not only dose dependent but also time-dependent. We therefore recommend monitoring the cumulative dose of daptomycin, which is a product of the dosage and length of therapy, rather than considering either variable separately. In our experience, clinicians should be cautious when the TCDD is ≥10 g, and particularly if it increases to ≥15 g, which can be easily attained after 2 weeks of treatment in patients receiving high doses.

Cases with DEP at the end of therapy had higher blood eosinophil counts and more often eosinophilia than controls, a fact that has been mainly reported previously [20,21]. Of interest, we noted a scenario in which severe DEP occurred shortly after a re-challenge with daptomycin, indicating that a drug hypersensitivity mechanism may be play. These cases presented with eosinophilia at the end of their previous course of daptomycin, a finding that was rarely observed in patients given a rechallenge without developing DEP. This clinical situation has been poorly reported to date [22], but it seems that eosinophilia during daptomycin therapy should prompt clinicians to consider avoiding further drug exposure.

The main limitations of the study are those inherent to the retrospective design. Generalizability is affected because patients were recruited from a single center and because the cohort mostly comprised elderly people with heterogeneous clinical presentations of osteoarticular infections. Also, unfortunately, our sample size of DEP cases was small to allow subgroup analyses or to design other comparative study. These factors must be factored when considering other heterogeneous populations. Irrespective of these shortcomings, however, we believe that our results provide information that can be led to improved management of daptomycin therapy. 

## 4. Materials and Methods

### 4.1. Study Design, Setting, and Inclusion/Exclusion Criteria

This retrospective cohort study was performed at the Bone and Joint Infection Unit of the Hospital Universitari Bellvitge between January 2014 and December 2018. We included all patients with osteoarticular infection (prosthetic joint infection, septic arthritis and osteomyelitis), aged ≥18 years, and treated at least with one dose of daptomycin because of empirical treatment or guided therapy addressed to Gram-positive microorganisms. Polymicrobial osteoarticular infections treated with daptomycin in combination with other antibiotics were also included. We excluded cases attended in our Bone and Joint Infection Unit that received daptomycin due to causes different than osteoarticular infections (i.e., catheter-related sepsis).

To identify risk factors for DEP, cases were divided into two groups: Those who developed DEP and those without DEP.

Written informed consent was considered unnecessary for the study, as it was a retrospective analysis of our clinical practice. Data of patients were anonymized for the purposes of this analysis. Confidential information of patients was protected according National and European normative. This manuscript has been revised for its publication by Research Ethics Committee of Bellvitge University Hospital (PR097/21).

### 4.2. Definitions and Clinical Data

All cases fulfilled the main diagnostic criteria for each osteoarticular infection, including those with prosthetic joint infection or osteoarthritis, with or without an orthopedic device.

The modified diagnostic criteria established by Philips et al. were used to define DEP [23], which required exposure to daptomycin with the following features: fever, dyspnea with increased oxygen requirement or requiring mechanical ventilation, new infiltrates on chest X-ray or computed tomography, and clinical improvement following daptomycin withdrawal. In accordance with these criteria, we did not require the previous pre-requisite of a bronchoalveolar lavage with >25% eosinophils.

Demographic, clinical, radiological and analytical data were collected for the included cases. Chronic heart failure, chronic pulmonary disease and chronic kidney disease were defined according to accepted criteria. The total cumulative dose of daptomycin (TCDD) was defined as daily dose of daptomycin × days of treatment; the result was expressed in grams (g).

### 4.3. Statistical Analysis

Data were analyzed using Stata software (version 16.0, Stata Corporation, College Station, TX, USA). Categorical variables are described by counts and percentages, while medians and interquartile ranges (IQRs) are used to summarize continuous variables.

Univariate analysis was performed to screen the risk factors for DEP, and logistic regression models were built to estimate unadjusted hazard ratios (HR). In all situations, *p*-values of <0.05 were considered to be statistically significant.

## 5. Conclusions

In conclusion, main factors associated with DEP were advanced age, high values of Charlson score, longer treatments and high total cumulative doses of daptomycin. Particularly, clinicians should take care in cases with cumulative doses greater than 10g, which can be achieved after 2 weeks of daptomycin therapy. In this high risk population and after the beginning of treatment, performing a chest-X ray is useful to identify DEP. Where eosinophilia has previously occurred with daptomycin exposure, further drug challenges should be considered with great care to minimize the risk of DEP.

## Figures and Tables

**Figure 1 antibiotics-10-00446-f001:**
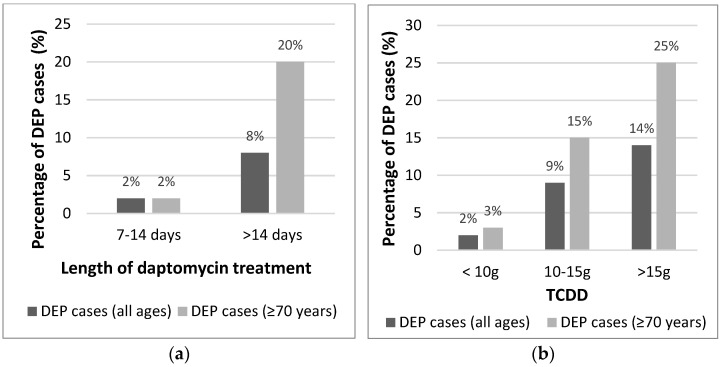
Percentage of Daptomycin-induced eosinophilic pneumonia (DEP) cases in the whole cohort and in those aged ≥70 years by (**a**) Length of therapy and by (**b**) The total cumulative dose of daptomycin (TCDD).

**Table 1 antibiotics-10-00446-t001:** Analysis of risk factors for daptomycin-induced eosinophilic pneumonia (DEP).

	Cases with DEP *n* = 11	Cases without DEP *n* = 218	HR (95% CI)	*p*-Value
Age (median, IQR)	77.4 (71.3–85.5)	69.7 (55.6–78.1)	1.06 (1.01–1.12)	0.042
<70 years			1	
≥70 years	10 (91)	108 (50)	10.19 (1.28–80.93)	0.028
Female	5 (45)	104 (48)	0.91 (0.27–3.08)	0.884
Comorbidities				
Charlson score (median, IQR)	5 (4–7)	4 (2–5)	1.31 (1.03–1.67)	0.031
Chronic heart disease	3 (30)	27 (12)	2.65 (0.66–10.62)	0.168
Chronic pulmonary disease	3 (30)	20 (9)	3.71 (0.91–15.12)	0.067
Chronic kidney disease	2 (20)	56 (26)	0.64 (0.13–3.07)	0.579
Analytical data (baseline)				
Creatinine (µmol/L)	62 (49–70)	72 (57–105)	0.98 (0.960–1.005)	0.134
Leucocytes (×10^9^ cells/L)	10.1 (7.3–11)	9.4 (7.2–12.3)	0.965 (0.829–1.122)	0.640
Eosinophils (cells/µL; median, IQR)	130 (30–230)	100 (30–240)	0.99 (0.996–1.003)	0.709
Analytical data (end of treatment)				
^2^Creatine kinase (mkat/L)	0.68 (0.31–0.89)	0.87 (0.54–1.85)	0.81 (0.498–1.316)	0.395
C-reactive protein (mg/L)	223 (120–315)	36 (17–83)	1.01 (1.007–1.018)	<0.001
Leucocytes (×10^9^ cells/L)	12.9 (9.5–15.4)	7.8 (6–9.8)	1.14 (1.035–1.258)	0.008
Eosinophils (cells/µL)	650 (520–1410)	220 (100–400)	1.01 (1.002–1.004)	<0.001
Daptomycin therapy				
Daily dose (mg; median, IQR)	700 (700–700)	700 (600–800)	1 (0.99–1.01)	0.719
Length (days; median, IQR)	19 (12–25)	7 (4–15)	1.08 (1.03–1.14)	0.005
≤14 days	3 (27)	162 (74)	1	
>14 days	8 (73)	56 (26)	7.71 (1.98–30.09)	0.003
^1^TCDD (g; median, IQR)	13.2 (8.4–17.5)	5.1 (2.4–11.2)	1.11 (1.03–1.19)	0.004
<10 g	3 (27)	155 (71)	1	
10–15 g	4 (36)	39 (18)	5.30 (1.14–24.66)	0.034
>15 g	4 (36)	24 (11)	8.61 (1.81–40.87)	0.007
Repeated exposure	2 (20)	23 (10)	1.88 (0.38–9.26)	0.435

Analytical data is presented as median, IQR. The remaining data are presented as *n* (%) unless otherwise noted. ^1^ TCDD (Total Cumulative Dose of Daptomycin; daily dose X days of treatment; The result was expressed in grams-g-) ^2^ Cases without DEP in which creatine kinase values were analyzed had a median of 12 days (IQR 6–19.5) of daptomycin therapy.

## Data Availability

The data presented in this study are available on request from the corresponding author (omurillo@bellvitgehospital.cat).

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
