# Peer review of "Risk Factors of Daptomycin-Induced Eosinophilic Pneumonia in a Population with Osteoarticular Infection"

_antibiotics, 2021, doi:10.3390/antibiotics10040446_

Round 1

Reviewer 1 Report

The manuscript is well structured and the supporting literature adequate. I have no suggestions for the authors.

Author Response

We apreciate your response, thanks.

Reviewer 2 Report

Soldevila-Boixader L.,et al aimed to to determine risk factors for Daptomycin-induced eosinophilic pneumonia (DEP).

The authors did a retrospective cohort study that was performed in a population with osteoarticular infection from January 2014 to December 2018.

To identify risk factors for DEP, cases were divided into two groups: 21 those who developed DEP and those without DEP.

In general, the study is well designed and well performed and does not need improvements.

Author Response

We appreciate your response, thanks.

Reviewer 3 Report

I read with interest the manuscript “Risk factors of daptomycin-induced eosinophilic pneumonia in a population with osteoarticular infection” in which the author/s describe clinical features of daptomycin-induced pneumonia.

This is not a new topic but worth to be investigated.  However some points must be addressed before the paper is acceptable for publication.

  • I recommend to move “Materials and Methods section” after the “Introduction” and before the “Results” section.

  • “Materials and methods” section line 155: it is reported that radiological data were collected but these data are not illustrated in the paper. It would be interesting to describe how many patients underwent chest X-ray or CT scan, with relative imaging findings.

  • Moreover I suggest to report if any radiological differences were present between the population at high for developing daptomycin-induced eosinophilic pneumonia compared to the low risk one.

  • Did the author/s recommend to perform a chest X-ray or a CT scan in the high risk population and at what timing after beginning therapy?

Author Response

Reviewer 3

I read with interest the manuscript “Risk factors of daptomycin-induced eosinophilic pneumonia in a population with osteoarticular infection” in which the author/s describe clinical features of daptomycin-induced pneumonia.

This is not a new topic but worth to be investigated.  However some points must be addressed before the paper is acceptable for publication.

We thank you for your kind comments; we have done some modifications in the manuscript according to your suggestions in order to improve it.

  • I recommend to move “Materials and Methods” section after the “Introduction” and before the “Results” section.

We thank you for your comment. We agree that the presentation of the manuscript could be rather unusual; however we have followed the instructions for the manuscript submission according to the journal website and followed their own template for this purpose.

  • “Materials and methods” section line 155: it is reported that radiological data were collected but these data are not illustrated in the paper. It would be interesting to describe how many patients underwent chest X-ray or CT scan, with relative imaging findings.

Thank you for your comment. We revised the radiological data with more detail. We provide information about how many patients underwent chest X-ray and the radiological findings in patients with DEP and the other patients in “Results section” (lines 48-59).

  • Moreover I suggest to report if any radiological differences were present between the population at high for developing daptomycin-induced eosinophilic pneumonia compared to the low risk one.

We agree with your comment. After the revision of radiological data we completed some new information and we modified the “Results” section adding a new paragraph including details on the differences between the radiological data in the different groups of the cohort (lines 48-59).

  • Did the author/s recommend to perform a chest X-ray or a CT scan in the high risk population and at what timing after beginning therapy?

We agree with this comment, too. We developed this part in the “Discussion” section and “Conclusions” section according with the radiological data analysis that we have illustrated before (lines 100-102).

Reviewer 4 Report

The authors summarize their retrospective observation on DAP. The paper is good structured and the reader can follow the aim and concern why this can be an important paper. Unfortunately, there are several issues which have to be  addressed before proceeding further.

1) the introduction section fails to summarize facts and guidelines which have been internationally published. There are few papers cited and in pubmed there are some important manuscripts. In addition, the exact molecular mechanism on the development must be presented. Please rewrite the introduction section by including guidelines and why daptomycin is still using, although it is known that DAP can be seen.

2) the inclusion and exclusion criteria are not well presented. Please state here a detailed list of criteria

3) the biochemical and other diagnostics including x-ray are missing. Please include more read out parameters for comparison analyses

4) a limitation section should be written in more detail as this is a single center retrospective study

5) for the discussion section, a list of pro’s and con’s should be presented

6) a guideline for readers should be discussed on why daptomycin should be or not used for this kind of patients

Author Response

Reviewer 4

The authors summarize their retrospective observation on DAP. The paper is good structured and the reader can follow the aim and concern why this can be an important paper. Unfortunately, there are several issues which have to be addressed before proceeding further.

We thank you for your kind comments; we have done some modifications in the manuscript according to your suggestions in order to improve it.

1) the introduction section fails to summarize facts and guidelines which have been internationally published. There are few papers cited and in pubmed there are some important manuscripts. In addition, the exact molecular mechanism on the development must be presented. Please rewrite the introduction section by including guidelines and why daptomycin is still using, although it is known that DAP can be seen.

We agree that the introduction could be improved. According to your comments we have rewritten the introduction section with information of the use of daptomycin in osteoarticular guidelines and we have explained with more detail the mechanism of daptomycin eosinophilic pneumonia (lines 31- 47).

2) the inclusion and exclusion criteria are not well presented. Please state here a detailed list of criteria

We agree that the inclusion and exclusion criteria could be improved. According to your comments, we have detailed the inclusion and exclusion criteria in order to provide clearer concepts. The modification is included in “Material and Methods” section (lines 139-144).

3) the biochemical and other diagnostics including x-ray are missing. Please include more read out parameters for comparison analyses

Thank you for that interesting comment. We agree with your comments which have been mentioned by the other reviewer, too. For this, we included information on chest x-ray and the radiological data of patients in “Results” section (lines 48-59) and we included some biochemical data in the comparative analysis (Table1).

4) a limitation section should be written in more detail as this is a single center retrospective study

According with that comment we have completed more accurately the part of limitations section (lines 130-135) that was already mentioned.

5) for the discussion section, a list of pro’s and con’s should be presented

6) a guideline for readers should be discussed on why daptomycin should be or not used for this kind of patients

We agree with you and we modified the “Discussion” section adding information about the pro’s and con’s of the use of daptomycin in the scenario of osteoarticular infection (lines 106-115).

Round 2

Reviewer 4 Report

The authors revised the manuscript appropiately.